# Statistical Copolymers of *n*-Butyl Vinyl Ether and 2-Chloroethyl Vinyl Ether via Metallocene-Mediated Cationic Polymerization. A Scaffold for the Synthesis of Graft Copolymers

**DOI:** 10.3390/polym11091510

**Published:** 2019-09-16

**Authors:** Stavros Zouganelis, Ioannis Choinopoulos, Ioannis Goulas, Marinos Pitsikalis

**Affiliations:** Industrial Chemistry Laboratory, Department of Chemistry, National and Kapodistrian University of Athens, Panepistimiopolis Zografou, 15771 Athens, Greece

**Keywords:** metallocene complexes, vinyl ethers, cationic polymerization, ring opening polymerization, statistical copolymers, graft copolymers, poly(ε-caprolactone), poly(l-lactide)

## Abstract

The cationic statistical copolymerization of *n*-butyl (be) and 2-chloroethyl vinyl ether (CEVE), is efficiently conducted using bis(η^5^-cyclopentadienyl)dimethyl zirconium (Cp_2_ZrMe_2_) in combination with tetrakis(pentafluorophenyl)borate dimethylanilinum salt [B(C_6_F_5_)_4_]^–^[Me_2_NHPh]^+^, as an initiation system. The reactivity ratios are calculated using both linear graphical and non-linear methods. Structural parameters of the copolymers are obtained by calculating the dyad sequence fractions and the mean sequence length, which are derived using the monomer reactivity ratios. The glass transition temperatures (*T*_g_) of the copolymers are measured by Differential Scanning Calorimetry (DSC), and the results are compared with predictions based on several theoretical models. The statistical copolymers are further employed as scaffolds for the synthesis of graft copolymers having poly(vinyl ether)s as a backbone and either poly(ε-caprolactone) (PCL) or poly(l-lactide) (PLLA) as side chains. Both the grafting “onto” and the grafting “from” methodologies are employed. The reaction sequence is monitored by Size Exclusion Chromatography (SEC), NMR and IR spectroscopies. The advantages and limitations of each approach are thoroughly examined.

## 1. Introduction

Poly(vinyl ethers) (PVEs) are considered one of the most valuable classes of polymeric materials, due both to their academic interest and their industrial applications. These include their use as adhesives, surface coatings, lubricants, greases, elastomers, melting compounds, fibers and films [1]. Their molecular weight, the nature of their side alkyl group, the nature of the initiator used for their polymerization, their stereospecifity and their crystallinity directly affect their properties. PVEs are produced as viscous sticky liquids, rubbery or brittle solids or even as waxy materials when they contain long alkyl groups [2]. Particularly, poly(2-chloroethyl vinyl ether) (PCEVE) can be employed as an X-Ray-beam resist since its chlorine substituents on the alkoxy side chains improve sensitivity and resolution. PCEVE is hydrophobic, elastomeric and can be employed as an intermediate for the synthesis of more complicated macromolecular structures [3].

PVEs can be prepared exclusively by the cationic polymerization of the corresponding vinyl ethers, due to the electron donating nature of the ether substituent. Several initiating systems, such as protonic acids (e.g., CF_3_SO_3_H, CF_3_COOH, HClO_4_) [4], metal halides (e.g., SnCl_4_, FeCl_3_, BF_3_) [5], halogenated metal alkyls (e.g., RAlCl_2_, R_2_AlCl, RMgX, R = alkyl, X = halogen), cation forming salts (Ph_3_C^+^SnCl_5_^−^, MeCO^+^ClO_4_^−^) [6], and halogens (e.g., I_2_, IBr) [7], have been employed for the polymerization of vinyl ethers. These systems, though, are not living and they suffer multiple termination, chain transfer and other side reactions.

The truly living cationic polymerization of vinyl ethers was first reported by Higashimura and Sawamato in 1983 with the introduction of the HI/I_2_ system [8]. Since this first report, several systems have been developed based on the stabilization of the growing cations [4,8]. This led to the synthesis of well-defined PVEs, with controlled molecular weights and narrow molecular weight distributions, end-functionalized PVEs, block copolymers, multi-arm stars and more complex macromolecular architectures based on PVEs.

PVEs have been polymerized using titanium, zirconium and hafnium complexes by Sudhakar [9] and Masuda [10], however, in these cases polymers of broad molecular distributions and in less than quantitative yields were obtained. Group 4 metallocene and half-metallocene complexes have been employed as homogenous polymerization catalysts in the presence of suitable cocatalysts, resulting in products with unique control over the molecular and structural characteristics [11]. They have been used efficiently to polymerize α-olefins [12], (meth)acrylates [13], lactones [14], lactides [15], styrene [16] and isocyanates [17,18,19] as well.

Ethyl vinyl ether (EVE), *n*-butyl vinyl ether (BVE) and isobutyl vinyl ether (IBVE) have been homopolymerized successfully by employing the system bis(*η*^5^-cyclopentadienyl) dimethylhafnium/tetrakis(pentafluorophenyl)borate dimethylanilinium salt Cp_2_HfMe_2_/[B(C_6_F_5_)_4_]^–^[Me_2_NHPh]^+^ in quantitative yields. Additionally, statistical copolymers of EVE and BVE have been synthesized successfully [20].

2-Chloroethyl vinyl ether (CEVE) has been polymerized by employing the HI/I_2_ system and a study of the reaction kinetics was carried out by Fontanille [21]. It also has been copolymerized successfully with methyl vinyl ether (MVE) using the 1-Diethoxyethane/Trimethylsilyl Iodide/Zirconium Iodide (DEE/TMSI/ZnI_2_) system [3]. A new approach to polymerize CEVE with maghnite clay has been introduced by Chabani M. et al. [22] resulting, however, in products with relatively broad molecular distributions.

During the present study, the zirconocene complex and the respective activating borate system Cp_2_ZrMe_2_/[B(C_6_F_5_)_4_]^–^[Me_2_NHPh]^+^ [20] is employed to statistically copolymerize CEVE with BVE. The monomer reactivity ratios are estimated using various graphical methods and a computer program. Structural parameters of the copolymers are obtained by calculating the dyad monomer sequence fractions and the mean sequence length. The thermal properties of the copolymers are studied by Differential Scanning Calorimetry (DSC), and are compared with those obtained from the corresponding homopolymers in the frame of various theoretical models that have been developed to predict the glass transition temperature of the statistical copolymers.

Branched polymers are among the most interesting macromolecular architectures. They are composed of a main polymer chain, the backbone, to which one or more side chains, the branches, are chemically connected through covalent bonds [23,24,25,26,27,28]. The synthesis of randomly branched graft copolymers can be accomplished through the following general approaches: (1) the “grafting onto”; (2) the “grafting from”; and (3) the macromonomer method (or “grafting through” method). The “grafting onto” method involves the use of a backbone chain containing functional groups X randomly distributed along the chain and branches having reactive chain ends Y. The coupling reaction between the X groups of the backbone and the Y groups of the end-reactive branches lead to the formation of graft copolymers. Demonstrated in the “grafting from” method, active sites are generated randomly along the backbone. These sites are capable of initiating the polymerization of a second monomer, leading to graft copolymers. The most commonly used method for the synthesis of graft copolymers, however, is the macromonomer method. Macromonomers are oligomeric or polymeric chains bearing a polymerizable end group [29,30,31,32,33,34]. Copolymerization of preformed macromonomers with another monomer yields graft copolymers. The poly(vinyl ether)’s statistical copolymers of this study are employed as scaffolds leading to the synthesis of graft copolymers, employing either the grafting onto or the grafting from methodology. These structures are composed of a poly(vinyl ether) backbone and either poly(l-lactide) (PLLA) or poly(ε-caprolactone) (PCL) side chains.

## 2. Materials and Methods

### 2.1. Materials

Toluene was dried over calcium hydride and oligostyrllithium, where it remained before distilling in a dry flask and storage in a sealed bottle inside a glovebox. Dichloromethane was dried over calcium hydride.

n-Butyl vinyl ether 98% (BVE) and 2-chloroethyl vinyl ether 99% (CEVE), (both purchased from Aldrich, Saint Luis, MO, USA), were dried over calcium hydride. ε-Caprolactone 99% (CL), (purchased from ACROS Chemicals, Gotëborg, Sweden), was dried over calcium hydride. L-Lactide 98% (LLA), (purchased from TCI Chemicals, Tokyo, Japan), was recrystallized in dry toluene (Grams of Lactide/mL of Toluene 1:1), under nitrogen, by heating to 80–90 °C to dissolve l-Lactide and then leaving it at room temperature overnight. The solvent then was removed carefully to another flask, always under an inert atmosphere, and the crystallized monomer was left to dry under vacuum for at least 6 h.

Bis(cyclopentadienyl)dimethylzirconium(IV) 97% (Cp_2_ZrMe_2_) (purchased from Aldrich) and *N*,*N*-dimethylanilinium tetrakis(pentafluorophenyl)borate 98% [B(C_6_F_5_)_4_]^–^[Me_2_NHPh]^+^,.(purchased from TCI, Tokyo, Japan), were used as received without further purification. Copper(I) Bromide and *N*,*N*,*N*′,*N*′′,*N*′′-Pentamethyldiethylenetriamine 99% (PMDETA) were degassed and stored in the glovebox for subsequent use.

Stannous Octoate 95% Sn(Oct)_2_ was dried over calcium hydride and then filtered inside the glovebox where it was stored in a previously dried bottle. Propargyl alcohol was dried over calcium hydride and then distilled, under vacuum, in a dry flask. Finally, it was transferred to a dry bottle inside the glovebox, where it was stored before use.

### 2.2. Copolymerization of BVE and CEVE

During a typical experiment, Cp_2_ZrMe_2_ (68 mg, 0.27 mmol) and [B(C_6_F_5_)_4_]^–^[Me_2_NHPh]^+^ (200 mg 0.25 mmol) were added to a 100 mL Schlenk dried flask. Depending on the molar feed ratio of the monomers, the required amount of toluene to maintain the 5M monomer concentration for 5 g in the solution was added (for BVE:CEVE 20:80/40:60/50:50/60:40/80:20 4.4 mL/4.2 mL/4.1 mL/4 mL/3.7 mL) and the solution was left under stirring for one hour at room temperature for the activation of the zirconocene complex.

The required amounts of monomers for the copolymerization were added in a second 100 mL Schlenk flask (for BVE:CEVE 20:80/40:60/50:50/60:40/80:20 1.2 mL:3.9 mL/2.5 mL:2.9 mL/3.1 mL:2.5 mL/3.8 mL:2.1 mL/5.1 mL:1 mL) in the glovebox. Subsequent to the completion of the activation both Schlenk flasks were left for 5–10 min in an ice bath and, after purging the Schlenk line with argon, the monomer mixture was added to the solution of the active complex using a dry syringe, under a constant argon atmosphere. The reaction flask was then sealed under argon and left in a thermostatted 0 °C ethylene glycol/water bath and was left to react for the required time (for 2 h or less for the study of reactivity ratios). The yields after 2 h of reaction were up to 33.3% and after 24 hrs for the 50:50 and 80:20 ratios were 70% and 85%, respectively. The reaction was quenched by adding piperidine and the polymer was precipitated in cold methanol under stirring, yielding a waxy solid. Methanol then was removed, and the polymer was re-dissolved in dichloromethane and filtered by a pore No. 4 glass filter, where a small quantity of celite was added to enhance the filtering process to remove the salts formed during the quenching of the reaction. The filtered dichloromethane solution was received in a previously weighed glass container and was left overnight in a fume hood for the evaporation of dichloromethane. Finally, the copolymer was dried in a vacuum oven at 40–50 °C for at least four hours to achieve constant weight. The product (PBVE-co-PCEVE) was then weighed and stored in the sealed container at 4 °C. The composition of the copolymer was calculated by ^1^H NMR spectroscopy (Varian, USA).

### 2.3. Transformation of Chlorine Groups to Azide Groups

During a typical experiment for a copolymer with 59:41 monomer ratio, PBVE–*co*–PCEVE (3 g,12 mmol of Cl), sodium azide (7.8 g 120 mmol) and *N*,*N*-Dimethyl formamide, DMF, (70 mL) were added to a flask and left to react for 48 h in a thermostatted 80 °C oil bath under intense stirring. Following the completion of the reaction, DMF was removed by vacuum distillation and the product was dissolved in dichloromethane. The polymer solution was filtered in a glass pore 4 filter, enhanced by Celite, and the product was dried in a vacuum oven at 40–50 °C for at least four hours. The product (PBVE–*co*–PAzEVE) then was stored in the sealed container at 4 °C. The reaction was monitored mainly by IR spectroscopy (Perkin Elmer, Bucks, England). Quantitative yields were obtained by ^1^H NMR analysis.

### 2.4. Transformation of Chlorine Groups to Acetoxy Groups

During a typical experiment for a copolymer with 57:43 monomer ratio, PBVE–*co*–PCEVE (4 g, 16.7 mmol of Cl), and sodium acetate (4.1 g, 50.1 mmol) and DMF (70 mL) were added to a flask and left to react for 72 h in a thermostatted 110 °C oil bath under vigorous stirring. Following the completion of the reaction, DMF was removed by vacuum distillation and the product was dissolved in dichloromethane. The polymer solution was filtered in a pore No. 4 glass filter, enhanced by celite, and the product was dried in a vacuum oven at 40–50 °C for at least four hours. The product (PBVE–*co*–PAcOEVE) was then stored in the sealed container at 4 °C. The reaction was monitored by ^1^H NMR spectroscopy. The conversions ranged from 71% to 91% by ^1^H NMR analysis.

### 2.5. Transformation of Acetoxy Groups to Hydroxide Groups

During a typical experiment, PBVE–*co*–PAcOEVE (4.4 g, 15.3 mmol in acetoxy moieties) was dissolved in 90 mL of acetone. Sixty mL of 20% *w*/*v* aqueous solution of sodium hydroxide were added in the beaker and the mixture was left under vigorous stirring for at least 18 h (overnight). The acetone phase (still retaining the red-brown colour of the polymer) was then separated from the colorless aqueous phase and was concentrated to 15–20 mL. The concentrated polymer solution was then precipitated dropwise into 200 mL of deionized water under stirring. Following the completion of the precipitation the pH was measured and drops of a 1M solution of hydrochloric acid were added until neutral pH was obtained. The beaker was left under stirring in a fume hood overnight to remove acetone for the better precipitation of the polymer and, the next day, the pH was measured again. When the pH was neutral the water was carefully separated from the product and the polymer was dissolved in dichloromethane, during which magnesium sulfate was added to remove excess water. Subsequent to leaving the solution to interact for 2–3 h, the solution was filtered and the polymer solution (PBVE–*co*–PHOEVE) was dried in a vacuum oven at 40–50 °C for at least four hours. Finally, the sample was stored in a sealed container at 4 °C. The reaction was monitored by ^1^H NMR spectroscopy.

### 2.6. Synthesis of Alkyne End-Functionalized PCL or PLLA

During a typical experiment, propargyl alcohol (115 μL/58 μL/29 μL 2 mmol/1 mmol/0.5 mmol, depending on the molecular weight of the polymer chain), Sn(Oct)_2_ (340 μL/170 μL/85 μL 1 mmol/0.5 mmol/0.25 mmol, depending on the molecular weight of the polymer chain), 6 g of CL (5.8 mL) or LLA and toluene (18 mL for Cl, 30 mL for LLA) were added to a 100 mL dried Schlenk flask in the glovebox. The flask was then sealed and placed in a preheated oil bath at 100 °C for CL for 1 h/2 h/4 h, and at 90 °C for LLA for 1.5 h/3 h/6 h, depending on the desired polymer molecular weight. Following the desired reaction time, the flask was cooled to room temperature in a water bath and the product was diluted with dichloromethane and precipitated in cold methanol. Subsequent to filtration, the white solid was collected and left to dry overnight under vacuum conditions. Finally, it was stored at 4 °C. The yield of the reaction was around 80%.

### 2.7. Grafting “onto” Method

During a typical experiment, PBVE–*co*–PAzEVE 59:41 (229 mg/114 mg/57 mg 0.89 mmol/0.44 mmol/0.22 mmol of N_3,_ depending on the PCL or PLLA chain length), 4 g of alkyne end-functionalized PCL or PLLA, copper (I) bromide (128 mg/64 mg/32 mg 0.89 mmol/0.44 mmol/0.22 mmol), PMDETA (186 μL/93 μL/46 μL 0.89 mmol/0.44 mmol/0.22 mmol) and 40 mL of dry dichloromethane were added to a dry Schlenk flask under an inert atmosphere. The reaction was left under stirring at room temperature for 8 days. The product was precipitated in cold methanol and filtered. Fractionation was performed to receive pure products. Regarding the PCL grafts, the system of solvent/non-solvent was toluene/methanol, while for PLLA chloroform/heptane was used. Finally, to remove the traces of copper trapped in the products, the polymers were dissolved in dichloromethane and, after addition of DOWEX acidic resin, were left overnight for stirring. The products then were precipitated once more and, after filtration, dried overnight under vacuum conditions. Finally, they were stored at 4 °C.

### 2.8. Grafting “from” Method

During a typical experiment, PBVE–*co*–PHOVE 57:39 (491 mg/245 mg/123 mg 2 mmol/1 mmol/0.5 mmol of OH, depending on the desired molecular weight of the polymer chain) was added to a 100 mL Schlenk flask and, after degassing, at least 5 mL of dry toluene was distilled to dissolve the polymer and then removed under vacuum three times for further drying of the macroinitiator. The Schlenk flask was inserted into the glovebox where Sn(Oct)_2_ (340 μL/170 μL/85 μL 1 mmol/0.5 mmol/0.25 mmol, depending on the desired molecular weight of the polymer chain), 6 g of CL (5.8 mL) or LLA and toluene (18 mL for CL, 30 mL for LLA) were added before sealing the flask and removing it again from the glovebox. The flask with the reaction mixture was left in a preheated oil bath at 100 °C for CL for 1 h/2 h/4 h and at 90 °C for LLA for 1.5 h/3 h/6 h, depending on the desired polymer chain length respectively. Following the suitable reaction time, the flask was cooled to room temperature in a water bath, the product was diluted with dichloromethane and precipitated in cold methanol. Subsequent to filtration, the solid polymer was collected and left to dry overnight under vacuum. Finally, it was stored at 4 °C. The yield of the reaction was at least 90% for PLLA and around 80% for PCL.

### 2.9. Characterization Techniques

Size exclusion chromatography (SEC) experiments were carried out using a modular instrument consisting of a Waters Model 510 pump, a Waters Model U6K sample injector, a Waters Model 401 differential refractometer, and a set of 4 μ-Styragel columns with a continuous porosity range from 10^6^ to 10^3^ Å. The columns were housed in an oven thermostatted at 40 °C. THF or CHCl_3_ were the carrier solvents at a flow rate of 1 mL/min. The instrument was calibrated with polystyrene standards.

The glass transition temperatures were obtained by differential scanning calorimetry (DSC) using a 2910 modulated DSC model from TA instruments. The samples were heated or cooled at a rate of 10 °C/min. The second heating results were obtained in all cases.

^1^H NMR spectra were recorded in chloroform-d at 30 °C with a Varian Unity Plus 300/54 NMR spectrometer.

IR spectra were recorded in a Perkin–Elmer Model Spectrum 100 FT-IR using KBr pellets.

## 3. Results

### 3.1. Synthesis of Statistical Copolymers

The cationic copolymerization of BVE with CEVE was conducted in toluene solutions at 0 °C (Scheme 1). The copolymerizations were allowed to proceed to relatively low conversions, satisfying the differential copolymerization equation. The produced copolymers were purified by repeated precipitations in cold methanol, where both monomers were soluble. The molecular characteristics of the samples are given in Table 1. The feed molar ratio of the two monomers was varied, as shown in Table 1. The samples were distinguished by the various feed molar ratios of the monomers, e.g., sample 20/80 indicates the copolymer for the synthesis of which 20% BVE and 80% CEVE was employed as the molar feed composition. The molecular weights were measured by SEC using a calibration curve constructed by polystyrene standards. It is obvious that the statistical copolymers have a relatively broad molecular weight distribution. This behavior is attributed to the presence of termination and chain transfer reactions. It was already reported in the literature that the cationic polymerization of vinyl ethers is susceptible mainly to β-hydrogen abstraction leading to terminated chains having end double bonds. This effect is more pronounced in higher molecular weight samples [20,35,36]. Another reason, which may be responsible for the relatively high polydispersity of the copolymers, is the temperature of the copolymerization reaction. The well controlled polymerization of CEVE via conventional cationic initiators was reported to take place at −30 °C to control the side reactions [3,37,38]. The rate of polymerization of CEVE is very low, however, so at this low temperature the copolymerization reaction led to products having a very low composition in CEVE. Therefore, to increase the rate of incorporation of CEVE monomer units to the copolymer chain, the copolymerization reaction was conducted at 0 °C. Using this relatively high temperature, it is reasonable to expect that termination and side reactions are more effective, leading to broader molecular weight distributions.

^1^H NMR spectroscopy was employed for the characterization of the samples. The calculation of the copolymer composition was based on the integration of the peaks of the methyl hydrogens of the BVE’s side groups at 0.9 ppm and the signals resonating at 3.2–3.7 ppm (methine and methylene protons of both monomer units). The NMR spectra are provided in Figure 1.

### 3.2. Monomer Reactivity Ratios and Statistical Analysis of the Copolymers

The monomer reactivity ratios were determined using the Finemann–Ross (FR), inverted Finemann–Ross (IFR), Kelen–Tüdos (KT) and extended Kelen–Tüdos (ext KT) graphical methods [39,40]. A computer program, called COPOINT, was also employed [41]. According to the Finemann–Ross method, the monomer reactivity ratios were obtained by the equation:G = Hr_BVE_ − r_CEVE_(1)
where the reactivity ratios, r_BVE_ and r_CEVE_ corresponded to the BVE and CEVE monomers, respectively. The parameters G and H are defined as follows:
G = X(Y − 1)/Y(2)
and
H = X^2^/Y(3)
with
with X = M_BVE_/M_CEVE_(4)
and
Y = dM_BVE_/dM_CEVE_(5)
M_BVE_ and M_BVE_ are the monomer molar compositions in feed and dM_BVE_ and dM_CEVE_ the copolymer molar compositions, measured by ^1^H NMR spectroscopy.

The inverted Finemann–Ross method is based on the equation:G/H = r_BVE_ − (1/H)r_CEVE_(6)

The plots of the G versus H values, and the G/H versus 1/H values, yield the reactivity ratios r_BVE_ and r_CEVE_ from the intercept and the slope of the graphs.

Alternatively, the reactivity ratios can be obtained using the Kelen–Tüdos method which is based on the equation:
η = (r_BVE_ + r_CEVE_/α)ξ − r_CEVE_/α(7)
where η and ξ are functions of the parameters G and H:
η = G/(α + H) and ξ = H/(α + H)(8)
and α a constant which is equal to (H_max_H_min_)^1/2^, H_max_, H_min_ being the maximum and the minimum H values, respectively, from the series of measurements. Taking the linear plot of η as a function of ξ, the values of η for ξ = 0 and ξ = 1 are used to calculate the reactivity ratios according to the equations:
ξ = 0 ⇒ η = −r_CEVE_/α and ξ = 1 ⇒ η = r_BVE_(9)
The characteristic of the K–T method is that it gives equal weight to all data points, thus it provides more realistic results than other graphical methods. Considering the marginal conversions obtained in this study, however the ext K–T is the preferred method. It is based on the same equations as the conventional method (equation 7, 8, 9), modifying the G and H as follows:
G = (Y − 1)/α(10)
and
H = Y/z^2^(11)
where
z = log(1 − ζ_A_)/log(1 − ζ_B_)(12)
ζ_B_ = W [(μ + X)/(μ + Y)](13)
ζ_A_ = (X/Y) ζ_B_(14)
μ is the ratio of the molecular weight of BVE to the molecular weight of CEVE and W is the conversion of the copolymerization reactions.

The copolymerization data for all systems are provided in Appendix A in the Appendix A Section. The graphical plots concerning the methods previously reported are given in Appendix A, whereas the reactivity ratios are summarized in Table 2. All plots for the different graphical methods were linear, thus indicating that these reactions followed conventional copolymerization kinetics and the reactivity of the active polymerization chain end is determined by the terminal monomer unit only.

The computer program, COPOINT, evaluated the copolymerization parameters using comonomer feed/copolymer composition data, as obtained from the copolymerization experiments, which were conducted up to finite monomer conversion. Theoretically, the mathematical treatment can be applied up to full monomer conversion, however, it is recommended the conversion not be more than 30 mol %. COPOINT numerically integrated a given copolymerization equation in its differential form. The copolymerization parameters were obtained by minimizing the sum of square differences between the measured and the calculated copolymer compositions.

It is clear that all methods provided similar results concerning the reactivity ratios for both monomers. According to the data obtained by COPOINT, r_BVE_ = 2.82 and r_CEVE_ = 0.57. These results imply that the rate of BVE incorporation into the copolymer structure is substantially higher than the rate of CEVE incorporation. This case is referred as a nonideal non-azeotropic copolymerization [42]. Found in the literature, conventional cationic polymerization has been employed for the synthesis of statistical copolymers containing CEVE monomer units and block copolymers with PCEVE polymer chains [43,44,45,46]. There are very few detailed studies leading to the calculation of the reactivity ratios, however. During a recent example describing the copolymerization of methyl vinyl ether, MVE and CEVE, it was found that r_MVE_ = 0.76 and r_CEVE_ = 0.46 [3]. This result confirms that CEVE is a less reactive monomer than MVE but, in this case, under the specific experimental conditions, a more random structure was observed. The main difference with the present study was the initiation system, meaning that the initiator played a crucial role for the elucidation of the monomers’ reactivities. A similar metallocene mediated cationic copolymerization was reported previously by our group, describing the copolymerization of ethyl vinyl ether and BVE [20]. During that case, EVE was much more reactive than BVE, since r_EVE_ = 1.74 and r_BVE_ = 0.50. The copolymerization was conducted at −10 °C in acetonitrile solutions employing Cp_2_HfMe_2_/[B(C_6_F_5_)_4_]^–^[Me_2_NHPh]^+^ as the initiating system.

The statistical distribution of the dyad monomer sequences M_BVE_-M_BVE_, M_CEVE_-M_CEVE_ and M_BVE_-M_CEVE_ were calculated using the method proposed by Igarashi [47]:(15)$$X\;=\;\varphi_{BVE}-\frac{2\varphi_{BVE}\left(1-\varphi_{BVE}\right)}{1+{\left[{\left(2\varphi_{BVE}-1\right)}^{2}+4r_{BVE}r_{CEVE}\varphi_{BVE}\left(1-\varphi_{BVE}\right)\right]}^{1/2}}$$
(16)$$Y\;=\;\frac{4\varphi_{BVE}\left(1-\varphi_{BVE}\right)}{1+{\left[{\left(2\varphi_{BVE}-1\right)}^{2}+4r_{BVE}r_{CEVE}\varphi_{BVE}\left(1-\varphi_{BVE}\right)\right]}^{1/2}}\;$$
(17)$$Z\;=\;\frac{4\varphi_{BVE}\left(1-\varphi_{BVE}\right)}{1+{\left[{\left(2\varphi_{BVE}-1\right)}^{2}+4r_{BVE}r_{CEVE}\varphi_{BVE}\left(1-\varphi_{BVE}\right)\right]}^{\frac{1}{2}}}\;$$
where *X*, *Y* and *Z* are the mole fractions of the M_BVE_-M_BVE_, M_CEVE_-M_CEVE_ and M_BVE_-M_CEVE_ dyads in the copolymer, respectively, and Φ_BVE_ the BVE mole fractions in the copolymer. Mean sequence lengths μ_EVE_ and μ_BVE_ also were calculated using the following equations [47]:
(18)$$\mu_{CEVE}\;=\;1+r_{CEVE}\frac{\left[M_{CEVE}\right]}{\left[M_{BVE}\right]}$$
(19)$$\mu_{BVE}\;=\;1+r_{BVE}\frac{\left[M_{BVE}\right]}{\left[M_{CEVE}\right]}$$

The data are summarized in Table 3 and the variation of the dyad fractions with the BVE mole fraction in the copolymers is displayed in Figure 2.

### 3.3. Glass Transition Temperature of the Statistical Copolymers

The *T*_g_ values of the PBVE and PCEVE homopolymers, along with the statistical copolymers prepared by metallocene mediated cationic copolymerization, were measured by DSC. The results are displayed in Table 4, whereas the DSC thermograms are given in Appendix A. The homopolymers were found to have *T*_g_ values equal to –53.44 and –22.15 °C for PBVE and PCEVE, respectively, in close agreement with the literature values [43]. Regarding the statistical copolymers, one transition was observed and the *T*_g_ values decreased upon increasing the BVE content.

The thermal properties of the copolymers, in principle, depend on the chemical structure of the monomer units, their composition and their monomer sequence distributions. Several theoretical equations have been employed in the past to describe how these parameters may affect the glass transition temperature of the copolymers to predict the Tg value at any copolymer composition.

The simplest equation describing the effect of composition on *T*_g_ is the Gibbs–Di Marzio equation [48]:
*T*_g_ = φ_BVE_Tg_BVE_ + φ_CEVE_Tg_CEVE_(20)
where φ_BVE_, φ_CEVE_ are the mole fractions of BVE and the comonomer CEVE, respectively, in the copolymer and *T*_gBVE_, *T*_gCEVE_ the glass transition temperatures of the two homopolymers, respectively.

A similar relationship was introduced by Fox [49]:
(21)$$\frac{1}{Tg}\;=\;\frac{w_{BVE}}{Tg_{BVE}}+\frac{w_{CEVE}}{Tg_{CEVE}}$$
where w_BVE_ and w_CEVE_ are the weight fractions of BVE and CEVE in the copolymer.

Based on the free volume concept, Johnston proposed the following equation [50]:
(22)$$\frac{1}{Tg}\;=\;\frac{w_{BVE}P_{BVE-BVE}}{Tg_{BVE-BVE}}+\frac{w_{CEVE}P_{CEVE-CEVE}}{Tg_{CEVE-CEVE}}+\frac{w_{BVE}P_{BVE-CEVE}+w_{CEVE}P_{CEVE-BVE}}{Tg_{BVE-CEVE}}$$

It is assumed that the BVE-BVE, CEVE-CEVE and CEVE-BVE or BVE-CEVE dyads have their own glass transition temperatures, *T*_gBVE-BVE_, *T*_gCEVE-CEVE_ and *T*_gBVE-CEVE_, respectively. *T*_gBVE-BVE_ and *T*_gCEVE-CEVE_ can be considered the glass transition temperatures for the respective homopolymers, whereas *T*_gBVE-CEVE_ is the glass transition temperature of the alternating copolymer P(BVE–*alt*–CEVE). W_i_ is the weight fraction of the i component and P_BVE-BVE_, P_CEVE-CEVE_, P_BVE-CEVE_ and P_CEVE-BVE_ are the probabilities of having various linkages. These probabilities can be calculated using the monomer reactivity ratios:
(23)$$P_{BVE-BVE}\;=\;\frac{r_{BVE}}{r_{BVE}+\frac{\left[M_{CEVE}\right]}{\left[M_{BVE}\right]}}$$
(24)$$\;P_{BVE-CEVE}\;=\;\frac{\left[M_{CEVE}\right]}{r_{BVE}\left[M_{BVE}\right]+\left[M_{CEVE}\right]}\;$$
(25)$$\;P_{CEVE-BVE}\;=\;\frac{\left[M_{BVE}\right]}{r_{CEVE}\left[M_{CEVE}\right]+\left[M_{BVE}\right]}\;$$
(26)$$\;P_{CEVE-CEVE}\;=\;\frac{r_{CEVE}\left[M_{CEVE}\right]}{r_{CEVE}\left[M_{CEVE}\right]+\left[M_{BVE}\right]}\;$$

Barton suggested the following equation [51]:
*T*_g_ = XTg_BVE-BVE_ + YTg_CEVE-CEVE_ + ZTg_BVE-CEVE_(27)
where *X*, *Y*, *Z* are the monomer dyad fractions (Equations (8)–(10)).

To apply these theories, it is necessary to know the glass transition temperature of the respective alternating copolymers, however, these data are not provided in the literature. Therefore, the linearized forms of the Johnston and Barton equations were used to obtain the *T*_gBVE-CEVE_ values. The plots are given in Figure 3 and Figure 4. It is obvious that straight lines passing through the origin were obtained. This is an indication that these theoretical methods can better predict the *T*_g_ values of statistical copolymers or, in other words, that the monomer sequence distribution is an important parameter affecting the *T*_g_ of the statistical copolymer. The *T*_gBVE-CEVE_ values calculated by the Johnston and Barton equations were similar, i.e., 249.38 and 248.72 K, respectively. The *T*_g_ values predicted from all the above mentioned methods are given in Table 4. It is evident that the predicted values from all methods fit reasonably well with the experimental results, however, the Barton method is in almost perfect agreement with these results. This conclusion indicates that the sequence distribution of the monomer units played a distinctive role in the elucidation of the *T*_g_ values of the copolymers and the expression of the copolymer composition in volume fractions, rather than in weight fractions, was more efficient.

### 3.4. Synthesis of Graft Copolymers via the Grafting “onto” Technique

Graft copolymers having poly(vinyl ether) as the backbone and either poly(ε-caprolactone) (PCL) or poly(l-lactide) (PLLA) side chains were prepared via the grafting “onto” methodology employing the statistical copolymers PBVE–*co*–PCEVE as a scaffold for this approach. The reaction sequence is given in Scheme 2 and involves three steps:

a. The transformation of the pendant chlorine groups of the statistical copolymer to azides

b. The synthesis of linear PCL and/or PLLA chains having alkyne groups at the one chain end, and

c. The linking reaction between the PVE backbone and the PCL or PLLA branches, under suitable experimental conditions.

PBVE–*co*–PCEVE statistical copolymers were synthesized, as previously described, but in quantitative conversion. Five copolymers of different monomer molar feed compositions were prepared. The samples are denoted as S50/50, S70/30, S59/41, S86/14 and S80/20, where S stands for the statistical copolymer and the following numbers indicate the molar feed ratio of BVE and CEVE. It was found that the final copolymer composition was equal to the stoichiometry, as a result of the quantitative conversion of the copolymerization. The molecular characteristics of the samples are given in Table 5. The molecular weights were relatively close to the stoichiometric values and the molecular weight distributions were broad, as was expected considering the discussion regarding the synthesis of the statistical copolymers. The chlorine moieties were then transformed to azides after reaction with an excess of NaN_3_. The reaction took place at 80 °C in DMF for 48 h. DMF is a relatively good solvent for the copolymers and is non-volatile. Therefore, at the rather high reaction temperature there was no loss of the solvent and, therefore, no need to use a condenser to collect the volatile solvent. The reaction was allowed to take place for 48 h to achieve quantitative conversions. Following the completion of the reaction, DMF was removed by vacuum distillation and the product was dissolved in dichloromethane. The polymer solution was filtered in a pore No. 4 glass filter, enhanced by Celite to remove the insoluble salts. SEC traces showed no obvious degradation or crosslinking side reactions during the post-polymerization reaction. A broadening of the molecular weight distribution was reported, however. This effect can be attributed to the partial adsorption of the polar azide groups to the crosslinked polymeric material of the SEC columns. The transformation of the pendant chlorines to the corresponding azides can be verified by IR spectroscopy. A characteristic example is given in Figure 5. The characteristic signals between 600 and 800 cm^−1^ of the chlorine groups disappeared after the azidation reaction and a new strong signal at 2100 cm^−1^, attributed to the azide groups, emerged. The statistical copolymers PBVE–*co*–PAzEVE are symbolized as the parent statistical copolymer with the addition of the letters Az, i.e., S50/50Az, S70/30Az, S50/50, S70/30, S59/41, S86/14 and S80/20and S80/20Az.

The second step involved the synthesis of linear PCL and PLLA chains with end-functional alkyne groups. This was achieved by polymerizing CL and LLA employing functional propargyl alcohol in the presence of stannous octoate (Sn(Oct)_2_) via a conventional Ring Opening Polymerization (ROP) reaction. It is well-known that semi-telechelic polymers with end-alkyne groups are prepared under these conditions. Samples of different molecular weights and of relatively narrow molecular weight distribution were obtained through this approach. To avoid transesterification and other side reactions during ROP and achieve quantitative end-group functionalization, the polymerization conversion was always lower than 80%. This was verified for the low molecular weight samples by ^1^H NMR spectroscopy by tracing the end-alkynyl groups and the adjacent methylene protons of the initiator and comparing this with the signals of the main polymer chain. The good agreement between the stoichiometric molecular weights and the molecular weights calculated by NMR confirmed the efficient synthesis of the end-functionalized polymers.

The final step for the synthesis of the desired PBVE–*g*–PCL and PBVE–*g*–PLLA graft copolymers involved the linking reaction of the PVE backbone and the PCL and PLLA side chains. This was conducted by the click reaction between the azide groups of the PVE backbone and the end-alkyne groups of either the PCL or PLLA side chains. The reaction was conducted in dichloromethane at room temperature in the presence of CuBr as the catalyst and PMDETA as the ligand. To achieve the highest linking efficiency, a small excess of the end-functionalized PCL and PLLA chains was employed and the linking was allowed to take place for 8 days. The reaction sequence was monitored by SEC. Characteristic examples from the synthesis of graft copolymers are shown in Figure 6. The molecular characteristics of the samples are provided in Table 5 and Table 6. The linking reaction was not quantitative, as was judged by SEC analysis. Considering the excess PCL or PLLA side chain and the SEC trace of the crude product after the linking reaction, it was concluded that the linking efficiency was higher in the case of PCL (about 80%) compared to PLLA (around 40%). This experimental behavior can be attributed to steric effects of the different polymer chains rather than the absence of the desired alkynyl end-group in PCL and PLLA chains. This conclusion can be further supported considering the monomer reactivity ratios of the statistical copolymer, which was actually the backbone of the graft copolymer. It was found that *r*_BVE_ was much higher than *r*_CEVE_. This result indicated that, during the synthesis, the statistical copolymer chain was initially rich in BVE and the incorporation of the CEVE units took place mainly at the latest parts of the copolymer chain. Therefore, the linking points for the grafting reaction mostly were accumulated at the one chain end of the copolymer and, consequently, steric problems were very pronounced. Fractionation using the toluene (solvent)/methanol (non-solvent) system was applied to purify the samples with PCL side chains, whereas chloroform (solvent)/heptane (non-solvent) was employed for the graft copolymers with PLLA side chains. The final products gave symmetric SEC traces and relatively narrow molecular weight distributions. ^1^H NMR spectroscopy was also employed as a tool to confirm the synthesis of the desired products. Characteristic examples are given in Figure 7.

### 3.5. Synthesis of Graft Copolymers via the Grafting “from” Technique

The grafting “onto” procedure, described above, was efficient for the synthesis of the desired products, however, the linking reaction was a time consuming process and the linking efficiency less than quantitative and, furthermore, it was expected to decrease upon increasing the molecular weight of the side chain and the number of the grafted available sites. Additionally, this approach required an extra step of purification, the fractionation, for the removal of the excess side chains. To overcome these shortcomings, the grafting “from” methodology also was applied for the synthesis of the same graft copolymers. This approach involved the following steps, as shown in Scheme 3:

a. The transformation of the pendant chlorine groups to the corresponding hydroxyl groups.

b. The employment of the available hydroxyl groups as initiation sites for the polymerization of CL or LLA.

The post-polymerization modification of the original statistical copolymer was conducted in a two-step procedure. Initially, the chlorine groups reacted with an excess of CH_3_COONa at high temperature (110 °C) to transform the halogens to the corresponding acetoxy groups. The reaction was conducted in DMF, which is a good and non-volatile solvent, for 72 h to achieve the highest conversion. The reaction was monitored by NMR spectroscopy and showed that the conversion of the chlorines to the corresponding acetoxy groups was very high (70% for the one sample and 91% for the other). SEC analysis of the product showed no obvious change to the PVE original chain. However, as in the case of the transformation of the chlorine groups to azides, the polydispersity of the polymer after the post-polymerization acetylation was slightly broader due to partial adsorption of the sample to the crosslinked polymer of the SEC columns. The acetylated copolymers PBVE-co-PAcOEVE are symbolized with the symbol of the precursor statistical copolymer followed by the letters Ac, e.g., S59/41Ac.

Subsequent basic hydrolysis resulted in the formation of the corresponding polymers where, finally, the chlorine groups were converted to hydroxyl groups. The copolymers PBVE-co-PHOEVE are symbolized in a similar manner, with the additional symbol HO, e.g., S59/41OH. The reaction took place overnight in acetone solutions using aqueous solutions of NaOH. NMR spectroscopy revealed that the hydrolysis was conducted in almost quantitative yields. Finally, the hydroxyl moieties served as initiation sites for the ROP of either CL or LLA in the presence of Sn(Oct)_2_ as a catalyst at high temperature (at 100 °C for the ROP of CL and at 90 °C for the ROP of LLA) in toluene solutions. The conversion of CL was at least 80% and for LLA at least 90%. The synthetic procedure was monitored by SEC and ^1^H NMR spectroscopy. Characteristic examples are given in Figure 8 and Figure 9. The small elution peak (4% of the crude product) in the SEC trace of Figure 8 is attributed to PLLA linear polymer. This was revealed after fractionation of the crude product to remove this trace and by ^1^H NMR analysis. The formation of this byproduct was attributed to the accidental presence of remaining reactive –OH groups in the system, obviously due to incomplete purification of the poly(vinyl ether) backbone. The other samples of the grafting “from” procedure did not show the presence of such byproduct. Considering the sensitivity of the NMR spectrophotometer, it seems that the vast majority of the available pendant hydroxyl groups were efficiently employed as initiating sites for the ROP of CL and LLA. Therefore, the grafting “from” methodology led to higher grafting efficiencies compared to the grafting “onto” approach, however, the molecular weight distributions of the products resulted via the grafting “from” methodology were broader than in the grafting “onto” approach. This is reasonable, since the rate of initiation of all the available hydroxyl groups along the PVE backbone was not equal due to the different local chemical environments at which they were placed. Therefore, the molecular weight distribution of the side chains was rather broad, leading, finally, to graft copolymers with broader distributions (Table 7).

The solid state and solution properties in both good and selective solvents will be presented in a forthcoming publication.

## 4. Conclusions

Statistical copolymers of *n*-butyl (BVE) and 2-chloroethyl vinyl ether (CEVE) were synthesized efficiently via metallocene-mediated cationic polymerization, employing bis(η^5^-cyclopentadienyl)dimethyl zirconium (Cp_2_ZrMe_2_) in combination with tetrakis (pentafluorophenyl) borate dimethylanilinum salt [B(C_6_F_5_)_4_]^–^[Me_2_NHPh]^+^ as the initiation system in toluene solutions at 0 ^°^C. The reactivity ratios were calculated using both linear graphical and non-linear methods. It was found that the reactivity ratio of BVE was much higher than that of CEVE, meaning that the rate of polymerization BVE was much higher than that of CEVE. Structural parameters of the copolymers were obtained by calculating the dyad sequence fractions and the mean sequence length, which were derived using the monomer reactivity ratios. These results indicated that the copolymer chains were rich in BVE monomer units at the initial stages of the copolymerization and CEVE was inserted at the chain mainly at the later stages of the copolymerization. The glass transition temperatures (*T*_g_) of the copolymers were measured by Differential Scanning Calorimetry (DSC) and the results were compared with predictions based on several theoretical models. The statistical copolymers were further employed as scaffolds for the synthesis of graft copolymers having poly(vinyl ether)s as a backbone and either poly(ε-caprolactone) (PCL) or poly(l-lactide) (PLLA) as side chains. Both the grafting “onto” and the grafting “from” methodologies were employed. The reaction sequence was monitored by Size Exclusion Chromatography (SEC), NMR and IR spectroscopies. The grafting “onto” approach led to products with predetermined molecular weights of the backbone and the side chains and with relatively narrow molecular weight distributions. Time consuming linking reactions were required, however, and the grafting efficiency was not very high. Conversely, the grafting “from” approach led to much higher grafting efficiencies but to products with broader molecular weight distributions.

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
