# Peer review of "Statistical Copolymers of n-Butyl Vinyl Ether and 2-Chloroethyl Vinyl Ether via Metallocene-Mediated Cationic Polymerization. A Scaffold for the Synthesis of Graft Copolymers"

_polymers, 2019, doi:10.3390/polym11091510_

Round 1

Reviewer 1 Report

This paper described the cationic statistical copolymerization behavior of n-butyl vinyl ether and 2-chloroethyl vinyl ether with a zirconocene catalyst system. The authors successfully determined the reactivity ratio for this copolymerization system using several different approaches. A value is added to this paper by employing the obtained statistical copolymers as the platform for the grafting onto and grafting from syntheses of brush copolymers with PLLA/PCL side chains. Given the thorough library of polymers as well as the detailed characterizations and discussions, the reviewer recommends this manuscript for publication in Polymers after minor revisions.

It is better to add the yield data in the experimental section of the copolymerization, azidation, acetylation, and click reaction.

What is the small elution peak appeared in the SEC trace of PBVE-co-(PEVE-g-PLLA) in Figure 8? Please explain about this in the text. In addition, the authors should recheck the SEC traces for the other grafting-from products to probe weather such small peak appeared or not.

In Schemes 2 and 3 and Figure 9b, the stereochemistry of the PLLA chemical structure need to be clarified.

Author Response

Reviewer #1

The yields of all the reactions were given in the text. Table 1 reports the yield in the copolymerization reactions. The azidation reaction was performed in a quantitative yield, as shown in Tables 5 and 6 (same composition in CEVE and AzEVE in the copolymers). The yields of the acetylation reactions were given in Table 7 (71% for sample S59/41 and 91% for sample S57/43). The yield of the click reaction is actually the grafting efficiency given in Tables 5 and 6. This information was also added in the experimental section, as the reviewer suggested.    

The small elution peak (4% of the crude product) in the SEC trace of Figure 8 is attributed to PLLA linear polymer. This was revealed after fractionation of the crude product to remove this trace and by NMR analysis. The formation of this byproduct was attributed to the accidental presence of remaining reactive –OH groups in the system, obviously due to incomplete purification of the poly(vinyl ether) backbone. The other samples of the grafting “from” procedure didn’t show the presence of such byproduct. A suitable comment was added in the text.

We didn’t examine in detail the stereochemistry of the PLLA structures, since there was reason to do this. Conventional Ring Opening Polymerization reactions were employed catalyzed by the well known catalyst Sn(Oct)2  (e.g. J. Pretula, S. Slomkowski, S. Penczek, Adv. Drug Deliv. Rev. 2016, 107, 3).

Reviewer 2 Report

Did authors recorded NMR for the crude mixtures of the co-polymers before precipitation. If not, that will be helpful to determine the % conversion for each monomer. This will also validate the computed reactivity ratios. Figure 9A. What solvent was used for nmr analysis. Should be mentioned in the caption. 9b: what nmr solvent was used?

Author Response

Reviewer #2

            The NMR spectra of the crude mixtures of the copolymers before precipitation were not recorded, since in this mixture we would have the signals of the copolymers, the remaining monomers and the reaction solvent. It would be very difficult to analyze such spectra and reach safe conclusions. Therefore, we only recorder the NMR spectra of the purified and dry copolymers.

            The NMR solvent was added in the figure captions, as indicated by the reviewer.